# Low-Rank Graph Neural Networks Inspired by the Weak-balance Theory in Social Networks

## Abstract

Graph Neural Networks (GNNs) have achieved state-of-the-art performance on node classification tasks by exploiting both the graph structures and node features. Generally, most existing GNNs depend on the implicit homophily assumption that nodes belonging to the same class are more likely to be connected. However, GNNs may fail to model heterophilious graphs where nodes with different labels tend to be linked, as shown in recent studies. To address this issue, we propose a generic GNN applicable to both homophilious and heterophilious graphs, namely Low-Rank Graph Neural Network (LRGNN). In detail, we aim at computing a coefficient matrix such that the sign of each coefficient reveals whether the corresponding two nodes belong to the same class, which is similar to the sign inference problem in Signed Social Networks. In this paper, we show that signed graphs are naturally generalized weakly-balanced for node classification tasks. Motivated by this observation, we propose to leverage low-rank matrix factorization (LRMF) to recover a coefficient matrix from a partially observed signed adjacency matrix. To effectively capture the node similarity, we further incorporate the low-rank representation (LRR) method. Our theoretical result shows that under the update rule of node representations, LRR obtained by solving a subspace clustering problem can recover the subspace structure of node representations. To solve the corresponding optimization problem, we utilize an iterative optimization algorithm with a convergence guarantee and develop a neural-style initialization manner that enables fast convergence. Finally, extensive experimental evaluation on both real-world and synthetic graphs has validated the superior performance of LRGNN over various state-of-the-art GNNs. In particular, LRGNN can offer clear performance gains in a scenario when the node features are not informative enough. Our code is available at https://anonymous.4open.science/r/lrgnn-4551/.

## 1 Introduction

Graphs (or networks) are ubiquitous in a variety of fields, such as social networks, biology, and chemistry. Many real-world networks follow the Homophily assumption, *i.e.*, linked nodes tend to share the same label or have similar features; while for graphs with heterophily, nodes with different labels are more likely to form a link. For example, many people tend to connect with people of the opposite sex in dating graphs. For graphs with homophily, Graph Neural Networks (GNNs) variants (Kipf & Welling, 2017; Hamilton et al., 2017; Velickovic et al., 2018) have achieved remarkable successes on various graph mining tasks. Among them, Graph Convolutional Network (GCN) (Kipf & Welling, 2017) and Graph Attention Networks (GATs) (Velickovic et al., 2018) are representative methods. However, the performance of GNNs deteriorates when learning on graphs with heterophily, in that the smoothing operation used in traditional GNNs tends to make representations of neighboring nodes similar, even though they have different labels.

Some designs (Zhu et al., 2020; Chien et al., 2021; Lim et al., 2021) have been proposed to enhance the representational power of GNNs under heterophilous scenarios (see Zheng et al. (2022) for a survey). Among them, high-pass filters are the most frequently used components since they can push away a node from its neighbors in the embedding space, which conforms to the characteristic of heterophily that nodes are generally dissimilar to their neighbors. High-pass filters are usually realized by negating the normalized adjacency matrix. In the spatial graph convolution domain, signed message passing (Yan et al., 2021; Bo et al., 2021) allows negative aggregation coefficients

so as to push away those neighboring heterophilious nodes. However, most existing methods have weaknesses that restrict their representational power. Spectral-based methods (Chien et al., 2021; Luan et al., 2021) combine high-pass filters with low-pass ones by linearly combining the outputs of intermediate layers. These methods fail to capture the node-level homophily ratio as they utilizes only one type of convolutional filters in each layer. Additionally, spatial-based methods (Bo et al., 2021; Yang et al., 2021) update the representation of each node by computing a learnable weighted combination of the representations of its neighbors and updating the aggregation coefficients based on the attention function used in Graph Attention Network (GAT) (Velickovic et al., 2018). As GAT computes a form of static attention and the ranking of the attention scores is unconditioned on the query node (Brody et al., 2022), this attention function is prone to produce uniform attention scores and cannot distinguish nodes of different classes when their distributions of features are of small difference (Fountoulakis et al., 2022).

In this paper, we address the challenges of generalizing GNNs to heterophilious graphs with the help of social theory developed from signed social networks (SSNs) whose positive links represent the friendship between two users while whose negative links represent enmity. Similarly, we call the graphs with negative edges as signed graphs. In SSNs, a practical theory called the weak balance theory (Davis, 1967) modifies the structural balance theory (Cartwright & Harary, 1956) by eliminating the patter "an enemy of my enemy is my friend", and keeping the following patterns of "an enemy of my friend is my enemy", "a friend of my friend is my friend", and "a friend of my enemy is my enemy". In the context of node classification task, we can view homophily and heterophily as friendship and enmity, respectively. It is easy to verify that signed graphs are weakly balanced when considering node classification tasks, e.g., a homophilious neighbor of my homophilious neighbor is also a homophilious node to me but a heterophilious neighbor of my heterophilious neighbor is not necessarily a homophilious node. The weak balance naturally leads to a global low-rank structure for the network, based on which the sign inference problem can be formulated as a low-rank matrix completion problem (LRMC) (Hsieh et al., 2012).

The weak-balance theory motivates us to apply low-rank approximation approaches to node classification on heterophilious graphs. Specifically, given a partially observed signed adjacency matrix (where negative edges are allowed), we aim to recover a coefficient matrix $\mathbf{Z}$ via low-rank approximation methods, such that a positive $\mathbf{Z}_{i,j}$ implies node $v_i$ and $v_j$ have the same label and the magnitude of $\mathbf{Z}_{i,j}$ represents the importance of node $v_j$ to $v_i$. Then we can use $\mathbf{Z}$ to update node representations by performing feature propagation in GNNs. Due to the polynomial time complexity of solving LRMC which is practically infeasible for large networks, we resort to the low-rank matrix factorization (LRMF) technique that is scalable to large graphs. The low-rank approximation of signed networks can achieve satisfactory or even exact recovery under certain assumptions as stated in Davenport & Romberg (2016). Furthermore, to better capture the similarity between node representations, we leverage the low-rank representation(LRR) (Liu et al., 2010) learning for recovering the underlying subspace structure on heterophilious graphs. Thus, we name the new derived GNN as Low Rank Graph Neural Network (LRGNN). It is important to note that the low-rank assumption was used to improve the defense over adversarial examples (Jin et al., 2020), it is not designed for heterophilious graph modeling and not implemented for signed graphs. To solve the corresponding non-convex optimization problem, we utilize the softImpute-ALS algorithm (Hastie et al., 2015) which minimizes the objective function by minimizing a surrogate function. Though LRGNN characterizes a node's representation using the linear combination of all the node representations, we can reduce the time complexity to linear by leveraging some tricks of matrix multiplication. Extensive experimental results on both real-world and synthetic datasets showed the superior performance and efficiency of LRGNN over state-of-the-art methods. We put all proofs and the discussion of related work in Appendix due to the space limitation.

## 2 PRELIMINARIES

**Notations**. Denote by $\mathcal{G} = (\mathcal{V}, \mathcal{E})$ a undirected graph, where $\mathcal{V}$ and $\mathcal{E}$ denote the node set and edge set, respectively. The nodes are described by a node feature matrix $\mathbf{X} \in \mathbb{R}^{n \times f}$, where $n$ and $f$ are the number of nodes and number of features per node, respectively. $\mathbf{Y} \in \mathbb{R}^{n \times c}$ is the node label matrix. The neighbor set of node $v_i$ is denoted by $\mathcal{N}_i$. We denote the node representation matrix in the $l$-th layer by $\mathbf{H}^{(l)}$. We let $\mathbf{A}$ denote the adjacency matrix that generally $\mathbf{A}_{i,j} = 1$ if $(i,j) \in \mathcal{E}$ and 0 otherwise; while in signed graphs (networks), we extend the values of adjacency matrix to

$\{-1, 0, 1\}$, where 1 corresponds to homophily (e.g., friendship), -1 corresponds to heterophily (e.g., enmity ), and 0 stands for unknown relationship. We call this generalized adjacency matrix as the signed adjacency matrix and denote it by $\tilde{\mathbf{A}}$. The Frobenius norm of a matrix is given by $\|\cdot\|_F^2$. $\mathbf{A}_{i,:}$ represents the $i$-th row of matrix $\mathbf{A}$.

We define the underlying complete graph and generalized $k$-weakly balanced graph as follows.

**Definition 1.** *For a signed graph, its underlying complete graph is given by adding all the missing edges to the graph, with appropriate signs, so that the resulting complete graph contains no positive links that connect two enemies (heterophilious nodes).*

**Definition 2.** *A signed graph is said to be generalized c-weakly balanced if the nodes can be divided into c groups such that in its underlying complete graph, all the within-group edges are positive and all the between-group edges are negative.*

Signed graphs are naturally generalized c-weakly balanced when considering node classification tasks with c classes. Figure 1 shows an example of a 3-weakly balanced graph. The following theorem reveals the low-rank structures of generalized weakly balanced graphs.

**Theorem 1.** *(Hsieh et al., 2012) The signed adjacency matrix of the underlying complete graph of a generalized c-weakly balanced graph is exactly of rank c if $c > 2$, and rank 1 if $c \leq 2$.*

Considering low-rank structures, it is natural to model the edge sign inference problem as an LRMC problem since LRMC can provide theoretical guarantees for exact recovery under some conditions. Given a partially observed signed matrix adjacency $\tilde{\mathbf{A}}$, the task of LRMC is to find the lowest-rank solution among all the feasible solutions:

Figure 1: An illustrative example of a 3-weakly balanced signed graph and its adjacency matrix of rank 3. Color corresponds to label.

$$\min \quad \text{rank}(\mathbf{Z}), \quad s.t. \quad \mathrm{P}_\Omega(\mathbf{Z}) = \mathrm{P}_\Omega(\tilde{\mathbf{A}}), \qquad (1)$$

where $\mathrm{P}_\Omega(\cdot)$ is an element-wise function defined as $\mathrm{P}_\Omega(\mathbf{M}_{i,j}) = \mathbf{M}_{i,j}$ if $(i,j) \in \mathcal{E}$ and 0 otherwise, for an arbitrary matrix $\mathbf{M} \in \mathbb{R}^{n \times n}$. Unfortunately, (1) is a NP-hard problem. An approximate optimization problem of (1) can be obtained using the convex relaxation

$$\min \quad \|\mathbf{Z}\|_*, \quad s.t. \quad \mathrm{P}_\Omega(\mathbf{Z}) = \mathrm{P}_\Omega(\tilde{\mathbf{A}}), \qquad (2)$$

where $\|\mathbf{Z}\|_*$ denotes the nuclear norm of $\mathbf{Z}$, which is the tightest convex relaxation of rank. Note that the values of entries of $\mathbf{Z}$ can be continuous. This optimization problem can be solved to yield the global optimal solution in polynomial time in many cases (Candès & Recht, 2012). We next discuss the conditions of perfect recovery. Let $\tau$ be the class imbalance defined as $\tau \equiv max\{n/n_i\}$, where $n$ is the number of nodes, $n_i$ the number of nodes of the $i$-th class. A surprising result involving LRMC is that by solving (2), perfect recovery from the observations is possible if certain assumptions hold.

**Theorem 2.** *(Hsieh et al., 2012) (Recovery Condition for Signed Networks) Suppose we observe edges $\tilde{\mathbf{A}}_{i,j}$, $(i,j) \in \Omega$, from a k-weakly balanced signed network $\mathbf{A}^*$. Also, suppose that the following assumptions hold: (1) k is bounded ($k = O(1)$), (2) the set of observed entries $\Omega$ is uniformly sampled, and (3) number of samples is sufficiently large, i.e., $|\Omega| \geq C\tau^4 n log^2 n$, where C is a constant.*

*Then $\mathbf{A}_*$ can be perfectly recovered by solving (2) with probability at least $1 - n^{-3}$.*

Although LRMC can conditionally provide exact recovery, solving (2) requires performing singular value decomposition on a potentially large matrix, which is computationally expensive. In practice, real-world graphs may have a large number of nodes. To make this idea practical, a fast algorithm is needed. To this end, we further relax the objective as

$$\min \quad \|\mathrm{P}_\Omega(\mathbf{Z} - \tilde{\mathbf{A}})\|_F^2, \quad s.t. \quad \text{rank}(\mathbf{Z}) \leq q, \qquad (3)$$

where $c \leq q \ll n$ is a parameter. The constraint on the rank of $\mathbf{Z}$ can be guaranteed by decomposing $\mathbf{Z}$ into the product of two low-rank matrices. This gives an LRMF problem.

$$\min_{\mathbf{U}, \mathbf{V} \in \mathbb{R}^{n \times q}} \quad \|\mathrm{P}_\Omega(\mathbf{U}\mathbf{V}^T - \tilde{\mathbf{A}})\|_F^2 + \lambda\Theta(\mathbf{U}, \mathbf{V}), \qquad (4)$$

where $\lambda$ is a parameter and $\Theta(\cdot, \cdot)$ a regularizer that encourages some desired properties in $\mathbf{U}, \mathbf{V}$ such as sparsity, non-negative, and orthogonality. Though (4) is a difficult non-convex problem with no known globally convergent algorithm, we can adopt alternating minimization based algorithms to obtain approximate yet good solutions. We note that matrix factorization (MF) algorithms predict the sign and magnitude of aggregation coefficient $(\mathbf{U}\mathbf{V}^T)_{i,j}$ only using the information from the observed entries in $\tilde{\mathbf{A}}$.

## 3 LOW RANK GRAPH NEURAL NETWORKS

In this section, we present the overall framework of Low Rank Graph Neural Networks and the corresponding optimization algorithm.

### 3.1 OVERALL FRAMEWORK

Here, we detail our model design. Inspired by Lim et al. (2021), we first apply MLPs to fuse feature matrix and adjacency matrix into a lower-dimensional matrix $\mathbf{H}^{(0)} \in \mathbb{R}^{n \times c}$.

$$\mathbf{H}^{(0)} = (1 - \mu)MLP_X(\mathbf{X}) + \mu MLP_A(\mathbf{A}), \tag{5}$$

where $0 < \mu < 1$ is the balance term. We here use adjacency matrix $\mathbf{A}$ to exploit the information of graph topology as MF-based methods will recover a complete graph from observations, where information of the original graph topology is lost.

For the updating of deeper layers, we introduce a hyper-parameter $\beta$ as motivated from APPNP (Klicpera et al., 2019) by keeping the term $\beta\mathbf{H}^{(0)}$ which is known to especially useful for learning on heterophilous graphs since it is not restricted by the homophily assumption. Once we have a coefficient matrix $\mathbf{U}_*^{(l)}\mathbf{V}_*^{(l)^T}$ for the $l$-th layer, the node representations for the $(l + 1)$-th layer are updated as

$$\mathbf{H}^{(l+1)} = (1 - \beta)\mathbf{U}_*^{(l)}\mathbf{V}_*^{(l)^T}\mathbf{H}^{(l)} + \beta\mathbf{H}^{(0)}, \tag{6}$$

where $0 < \beta < 1$, $\mathbf{U}_*^{(l)}, \mathbf{V}_*^{(l)} \in \mathbb{R}^{n \times q}$ are derived by minimizing the following objective function

$$F(\mathbf{U}^{(l)}, \mathbf{V}^{(l)}) = \|\mathbf{H}^{(l)} - (1 - \beta)\mathbf{U}^{(l)}\mathbf{V}^{(l)^T}\mathbf{H}^{(l)} - \beta\mathbf{H}^{(0)}\|_F^2 + \gamma\|\mathrm{P}_\Omega(\mathbf{U}^{(l)}\mathbf{V}^{(l)^T} - \tilde{\mathbf{A}})\|_F^2, \tag{7}$$

where $\gamma$ weights the importance of the MF term. The first term is the propagation term which enables the aggregation coefficients to capture the similarity of node representations via involving matrix $\mathbf{H}^{(l)}$ and $\mathbf{H}^{(0)}$. It is important to note that the first term can be derived from the low-rank representation learning of the coefficient matrix (as will described in Section 4). This term is also closely related to GloGNN (Li et al., 2022) and the difference will be discussed in Section 4. The second term is the MF term that aims to recover the missing edges from observed ones. Therefore, our method captures node correlations and derives aggregation coefficients considering both the observed edges and node representations.

There are still several problems we need to address. First, the signed adjacency matrix $\tilde{\mathbf{A}}$ is not available. Second, optimizing (7) is not trivial due to the element-wise function $\mathrm{P}_\Omega(\cdot)$ and the propagation term. Finally, it is important to find a good starting point for $\mathbf{U}$ and $\mathbf{V}$ so as to converge within a small number of iterations.

To address the first problem, we can use any off-the-shelf neural network classifier to generate pseudo labels. We also exploit the known node labels in training set $\mathcal{T}_\mathcal{V}$ and the label matrix $\mathbf{Y}$. Similar to Zhu et al. (2021), we generate the pseudo labels as follows,

$$\bar{\mathbf{Y}} = \mathbf{O} \odot \mathbf{Y} + (1 - \mathbf{O}) \odot \hat{\mathbf{Y}}, \quad \hat{\mathbf{Y}} = \mathrm{softmax}(\mathbf{P}), \quad \mathbf{P} = f_{NN}(\mathbf{X}), \tag{8}$$

where $f_{NN}(\cdot)$ denotes a trained neural network, $\odot$ the Hadamard product, and $\mathbf{O}_{i,:} = \mathbf{1}$ if $i \in \mathcal{T}_\mathcal{V}$, $\mathbf{0}$ otherwise. The signed adjacency matrix is defined as $\tilde{\mathbf{A}}_{i,j} = \langle \bar{\mathbf{Y}}_{i,:}, \bar{\mathbf{Y}}_{j,:} \rangle - \delta$ if $(i, j) \in \mathcal{E}$ and $0$ otherwise. Here, $0 < \delta < 1$ is a parameter to control the ratio of negative edge weights. $\langle \cdot, \cdot \rangle$ denotes the inner product of two vectors. We note that one can use stored pseudo labels generated by any model. In our experiments, we use simple models including GCN and MLP for fairness.

## 3.2 SOFTIMPUTE ALTERNATING LEAST SQUARES ALGORITHM

Practical solutions to LRMF fall into two main camps, *i.e.*, Alternating Least Squares algorithm (ALS) (Wiberg, 1976; Shum et al., 1995; Huynh et al., 2003) and Newton methods (Buchanan & Fitzgibbon, 2005; Okatani & Deguchi, 2007; Chen, 2008). We refer the interested reader to Davenport & Romberg (2016) for a survey. Unfortunately, both these two methods are difficult to handle additional constraints. The propagation term hinders us to optimize (7) by ALS or Newton methods. To tackle this issue, we make use of the softImpute-ALS algorithm (Hastie et al., 2015). Consider that we have current estimates $\bar{\mathbf{U}}^{(l)}$ and $\bar{\mathbf{V}}^{(l)}$, and we now wish to derive a new $\tilde{\mathbf{U}}^{(l)}$ that minimize the objective. We first introduce the following surrogate function for deriving $\tilde{\mathbf{U}}^{(l)}$.

$$S_U(\mathbf{Z}_U|\bar{\mathbf{U}}^{(l)}, \bar{\mathbf{V}}^{(l)}) = \|\mathbf{H}^{(l)} - (1-\beta)\mathbf{Z}_U\bar{\mathbf{V}}^{(l)^T}\mathbf{H}^{(l)} - \beta\mathbf{H}^{(0)}\|_F^2 + $$
$$\gamma\|\mathrm{P}_\Omega(\mathbf{Z}_U\bar{\mathbf{V}}^{(l)^T} - \tilde{\mathbf{A}}) + \mathrm{P}_\phi(\mathbf{Z}_U\bar{\mathbf{V}}^{(l)^T} - \bar{\mathbf{U}}^{(l)}\bar{\mathbf{V}}^{(l)^T})\|_F^2 \quad (9)$$

where $\mathrm{P}_\phi(\cdot)$ is an element-wise function and defined as $\mathrm{P}_\phi(\mathbf{M}_{i,j}) = \mathbf{M}_{i,j}$ if $(i,j) \notin \mathcal{E}$ and 0 otherwise. Then $\tilde{\mathbf{U}}^{(l)}$ can be obtained by minimizing the surrogate function over $\mathbf{Z}_U$, *i.e.*, $\tilde{\mathbf{U}}^{(l)} = \underset{\mathbf{Z}_U \in \mathbb{R}^{n \times q}}{argmin} \; S_U(\mathbf{Z}_U|\bar{\mathbf{U}}^{(l)}, \bar{\mathbf{V}}^{(l)})$. The closed-form solution is (see Appendix A.1).

$$\tilde{\mathbf{U}}^{(l)} = [\gamma\hat{\mathbf{A}}\bar{\mathbf{V}}^{(l)} + (1-\beta)\mathbf{H}^{(l)}\mathbf{H}^{(l)^T}\bar{\mathbf{V}}^{(l)} - \beta(1-\beta)\mathbf{H}^{(0)}\mathbf{H}^{(l)^T}\bar{\mathbf{V}}^{(l)}] \cdot$$
$$[\gamma\bar{\mathbf{V}}^{(l)^T}\bar{\mathbf{V}}^{(l)} + (1-\beta)^2\bar{\mathbf{V}}^{(l)^T}\mathbf{H}^{(l)}\mathbf{H}^{(l)^T}\bar{\mathbf{V}}^{(l)}]^{-1}, \quad (10)$$

where $\hat{\mathbf{A}} = P_\Omega(\tilde{\mathbf{A}}) + P_\phi(\bar{\mathbf{U}}^{(l)}\bar{\mathbf{V}}^{(l)^T})$. Similarly, we define a surrogate function for deriving $\tilde{\mathbf{V}}^{(l)}$ .

$$S_V(\mathbf{Z}_V|\tilde{\mathbf{U}}^{(l)}, \bar{\mathbf{V}}^{(l)}) = \|\mathbf{H}^{(l)} - (1-\beta)\tilde{\mathbf{U}}^{(l)}\mathbf{Z}_V^T\mathbf{H}^{(l)} - \beta\mathbf{H}^{(0)}\|_F^2 + $$
$$\gamma\|\mathrm{P}_\Omega(\tilde{\mathbf{U}}^{(l)}\mathbf{Z}_V^T - \tilde{\mathbf{A}}) + \mathrm{P}_\phi(\tilde{\mathbf{U}}^{(l)}\mathbf{Z}_V^T - \tilde{\mathbf{U}}^{(l)}\bar{\mathbf{V}}^{(l)^T})\|_F^2 \quad (11)$$

We obtain $\tilde{\mathbf{V}}^{(l)}$ by minimizing the above surrogate function over $\mathbf{Z}_V$. The closed-form solution is

$$\tilde{\mathbf{V}}^{(l)} = [\gamma\mathbf{I}_n + (1-\beta)^2\mathbf{H}^{(l)}\mathbf{H}^{(l)^T}]^{-1} \cdot$$
$$[\gamma\hat{\mathbf{A}}^T\tilde{\mathbf{U}}^{(l)} + (1-\beta)\mathbf{H}^{(l)}\mathbf{H}^{(l)^T}\tilde{\mathbf{U}}^{(l)} - \beta(1-\beta)\mathbf{H}^{(l)}\mathbf{H}^{(0)^T}\tilde{\mathbf{U}}^{(l)}] \cdot [\tilde{\mathbf{U}}^{(l)^T}\tilde{\mathbf{U}}^{(l)}]^{-1} \quad (12)$$

Here $\hat{\mathbf{A}} = P_\Omega(\tilde{\mathbf{A}}) + P_\phi(\tilde{\mathbf{U}}^{(l)}\bar{\mathbf{V}}^{(l)^T})$. We establish the following result to justify the design of the surrogate functions.

**Theorem 3.** *(Correctness) The objective function (7) is non-increasing under the update rules (10) and (12),*

$$F(\tilde{\mathbf{U}}^{(l)}, \tilde{\mathbf{V}}^{(l)}) \le F(\tilde{\mathbf{U}}^{(l)}, \bar{\mathbf{V}}^{(l)}) \le F(\bar{\mathbf{U}}^{(l)}, \bar{\mathbf{V}}^{(l)}) \quad (13)$$

**Remark 1.** *One can calculate $\mathbf{H}^{(l+1)}$ in a time complexity linear to the number of edges using some tricks of matrix multiplication. See appendix A.3 for a detailed discussion.*

Most traditional approaches randomly initialize $\mathbf{U}$ and $\mathbf{V}$. In this paper, we employ a more neural-style initialization manner.

$$\mathbf{U}_{init}^{(l)} = \underset{\mathbf{U}}{argmin} \quad \|\tilde{\mathbf{A}} - \mathbf{U}\mathbf{V}_{init}^{(l)}\|_2^2 = (\tilde{\mathbf{A}}\mathbf{V}_{init}^{(l)})(\mathbf{V}_{init}^{(l)^T}\mathbf{V}_{init}^{(l)})^{-1}, \quad \mathbf{V}_{init}^{(l)} = f_{init}(\mathbf{H}^{(0)}) \quad (14)$$

Here $f_{init}(\cdot)$ denotes a fully-connected layer or graph convolution layer, depending on the homophily ratio. We empirically found that this initialization can provide better results within a few iterations for updates. The pseudocode of LRGNN can be found in Appendix A.2.

## 4 PLACING LRGNN IN THE CONTEXT OF SUBSPACE CLUSTERING

In this section, we show that the propagation term can be derived by performing subspace clustering using low-rank representation (LRR) (Liu et al., 2010). We consider the subspace clustering problem. Given a set of data samples (each sample is associated with a vector) drawn from a union of linear subspaces, the goal of subspace clustering is to group the samples into their respective subspaces. Spectral-type methods first learn a coefficient matrix from the given data, then the clustering

results can be obtained by applying spectral clustering methods to the coefficient matrix. Suppose we have data matrix $\mathbf{X} \in \mathbb{R}^{n \times d}$ at hand, we want to find a coefficient matrix such that each of data samples can be represented by the linear combination of the basis in a "dictionary" $\mathbf{D} \in \mathbb{R}^{n \times d}$ as $\mathbf{X} = \mathbf{ZD}$, where $\mathbf{Z}_{i,j}$ describes the affinity between the $i$-th sample and $j$-th sample and $\mathbf{Z}_{i,:}$ is the representation of the $i$-th sample. LRR aims at finding the lowest-rank representation among all the candidates.

LRR is robust to noise and outliers in the subspace clustering problem (Liu et al., 2010). Latent Low-Rank Representation (LatLRR) (Liu & Yan, 2011) further improves LRR with hidden data. We consider a variant of LatLRR,

$$\min \quad \|\mathbf{Z}\|_* + \lambda \|\mathbf{Z}\|_F^2, \quad s.t. \quad \mathbf{X} = [\mathbf{Z}||\mathbf{I}_n][(1-\beta)\mathbf{X}^T||\beta\mathbf{X}_H^T]^T = (1-\beta)\mathbf{ZX} + \beta\mathbf{X}_H, \quad (15)$$

where $\mathbf{X}_H \in \mathbb{R}^{n \times d}$ is the hidden data matrix and $\beta$ a parameter to weight the importance of hidden data. Back to our problem, now we wish to perform subspace clustering on $\mathbf{H}^{(l)}$. When we consider the graphs with heterophily, it is advisable to use the initial node representations $\mathbf{H}^{(0)}$ as hidden data since it is not restricted by the homophily assumption. By appropriately replacing the symbols, we obtain our final optimization problem

$$\min \quad \|\mathbf{Z}\|_* + \lambda \|\mathbf{Z}\|_F^2, \quad s.t. \quad \mathbf{H}^{(l)} = (1-\beta)\mathbf{ZH}^{(l)} + \beta\mathbf{H}^{(0)} \quad (16)$$

The low-rank structure induced by the nuclear norm regularization on $\mathbf{Z}$ is of great significance. We have the following result.

**Theorem 4.** *Assume that the row vectors (node representations) of $\mathbf{H}^{(0)}$ are drawn from a union of independent subspaces $\{S_i\}_{i=1}^c$. Also Assume that the update rule of node representation matrix is*

$$\mathbf{H}^{(l+1)} = (1-\beta)\mathbf{Z}^{(l)}\mathbf{H}^{(l)} + \beta\mathbf{H}^{(0)}, \quad (17)$$

*where $\mathbf{Z}^{(l)}$ is an optimal solution (assume it exists) to the following optimization problem*

$$\min \quad \|\mathbf{Z}\|_* + \lambda \|\mathbf{Z}\|_F^2, \quad s.t. \quad \mathbf{H}^{(l)} = (1-\beta)\mathbf{ZH}^{(l)} + \beta\mathbf{H}^{(0)}, \quad (18)$$

*where $\lambda > 0$, $\beta > 0$. Then for any node-pair (denoted by $v_i$ and $v_j$) that belong to different subspaces, we have $\mathbf{Z}_{i,j}^{(l)} = 0$, $\forall l \geq 0$.*

When the number of clusters is known (e.g., $c$) and the number of columns in $\mathbf{U}$ (i.e., $q$) is sufficiently larger than $c$, the nuclear norm minimization problem can be equivalently replaced by the low-rank decomposition (Cabral et al., 2013). We therefore have the following optimization problem

$$\min_{\mathbf{U}^{(l)}, \mathbf{V}^{(l)} \in \mathbb{R}^{n \times q}} \|\mathbf{H}^{(l)} - (1-\beta)\mathbf{U}^{(l)}\mathbf{V}^{(l)^T}\mathbf{H}^{(l)} - \beta\mathbf{H}^{(0)}\|_F^2 + \lambda'\|\mathbf{U}^{(l)}\mathbf{V}^{(l)^T}\|_F^2, \quad s.t. \quad q \geq c \quad (19)$$

The first term is exactly the propagation term in (7). Our numerical simulation experiments show that LRR obtained by solving (19) can reveal the membership of the samples: within-subspace elements are dense, and the between-subspace elements are sparse. We here include $\lambda'\|\mathbf{U}^{(l)}\mathbf{V}^{(l)^T}\|_F^2$ for theoretical completeness. In our simulations, as $\lambda'$ increases, within-subspace elements get closer to zeros, as shown in Figure 2. Therefore, our objective function excludes this regularization term. The detailed settings and more simulation results are provided in Appendix A.6. In summary, our objective function contains two parts. The propagation term is motivated by the LRR method for subspace clustering. The second MF term is inspired by the low-rank structures of the weakly-balanced graphs.

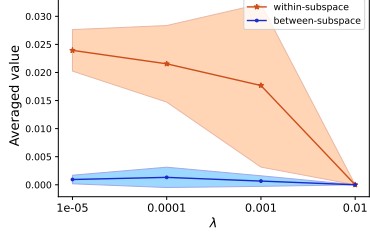

Figure 2: Numerical simulation result. The low-rank representation is obtained by solving (19). The shaded region indicates a 95% confidence interval.

**Discussion**. A closely related work is the GloGNN (Li et al., 2022), which defines the coefficient matrix as

$$\mathbf{Z}_*^{(l)} = \operatorname*{argmin}_{\mathbf{Z}^{(l)} \in \mathbb{R}^{n \times n}} \|\mathbf{H}^{(l)} - (1-\beta)\mathbf{Z}^{(l)}\mathbf{H}^{(l)} - \beta\mathbf{H}^{(0)}\|_F^2 + \gamma\|\mathbf{Z}^{(l)} - \mathbf{A}_{\text{GCN}}\|_F^2 + \lambda\|\mathbf{Z}^{(l)}\|_F^2 \quad (20)$$

| | Texas | Wiscon. | Cornell | Actor | Squirrel | Chamel. | Cora | Citeseer | Pubmed |
|---|---|---|---|---|---|---|---|---|---|
| MLP | 80.81±4.75 | 85.29±3.31 | 81.89±6.40 | 36.53±0.70 | 28.77±1.56 | 46.21±2.99 | 75.69±2.00 | 74.02±1.90 | 87.16±0.37 |
| GCN | 55.14±5.16 | 51.76±3.06 | 60.54±5.30 | 27.32±1.10 | 53.43±2.01 | 64.82±2.24 | 86.98±1.27 | 76.50±1.36 | 88.42±0.50 |
| GAT | 52.16±6.63 | 49.41±4.09 | 61.89±5.05 | 27.44±0.89 | 40.72±1.55 | 60.26±2.50 | 87.30±1.10 | 76.55±1.23 | 86.33±0.48 |
| MixHop | 77.84±7.73 | 75.88±4.90 | 73.51±6.34 | 32.22±2.34 | 43.80±1.48 | 60.50±2.53 | 87.61±0.85 | 76.26±1.33 | 85.31±0.61 |
| H$_2$GCN | 84.86±7.23 | 87.65±4.98 | 82.70±5.28 | 35.70±1.00 | 36.48±1.86 | 60.11±2.15 | 87.87±1.20 | 77.11±1.57 | 89.49±0.38 |
| GPR-GNN | 78.38±4.36 | 82.94±4.21 | 80.27±8.11 | 34.63±1.22 | 31.61±1.24 | 46.58±1.71 | 87.95±1.18 | 77.13±1.67 | 87.54±0.38 |
| WRGAT | 83.62±5.50 | 86.98±3.78 | 81.62±3.90 | 36.53±0.77 | 48.85±0.78 | 65.24±0.87 | 88.20±2.26 | 76.81±1.89 | 88.52±0.92 |
| GloGNN++ | 84.05±4.90 | 88.04±3.22 | 85.95±5.10 | **37.70±1.40** | 57.88±1.76 | 71.21±1.84 | **88.33±1.09** | 77.22±1.78 | 89.24±0.39 |
| GGCN | 84.86±4.55 | 86.86±3.29 | 85.68±6.63 | 37.54±1.56 | 55.17±1.58 | 71.14±1.84 | 87.95±1.05 | 77.14±1.45 | 89.15±0.37 |
| ACM-GCN | 87.84±4.40 | **88.43±3.22** | 85.14±6.07 | 36.28±1.09 | 54.40±1.88 | 66.93±1.85 | 87.91±0.95 | 77.32±1.70 | **90.00±0.52** |
| LINKX | 74.60±8.37 | 75.49±5.72 | 77.84±5.81 | 36.10±1.55 | 61.81±1.80 | 68.42±1.38 | 84.64±1.13 | 73.19±0.99 | 87.86±0.77 |
| **LRGNN** | **89.19±4.49** | 88.23±3.04 | **86.22±6.10** | 37.10±2.12 | **74.51±1.90** | **78.93±1.23** | 88.23±1.03 | **77.46±1.31** | 89.60±0.54 |

Table 1: Node classification results on 9 real-world benchmark datasets. The results we report are the averages associated with standard deviations over 10 trials. We highlight the best results in bold and the runner-up results with underlines.

Notice that although the first term in Problem (19) has a similar form as the first term in Equation (20), they are derived from different perspectives. We here point out several major differences between our LRGNN and GloGNN. First and most importantly, the coefficient matrix in LRGNN is explicitly modeled as a low-rank matrix. The propagation term in GloGNN is only the vanilla subspace clustering augmented with hidden data, neither sparse nor low-rank is ensured. However, LRR is shown to offer both theoretical and empirical benefits to subspace clustering. Moreover, the weak balance theory developed from signed social networks reveals the low-rank structures of the underlying complete graphs. Second, LRGNN aims at recovering the missing edge weights via matrix factorization while GloGNN encourages the missing edges to be near zeros by including a term $\|\mathbf{Z}^{(l)} - \mathbf{A}_{GCN}\|_F^2$. On the contrary, the MF term in LRGNN does not impose restrictions on the unobserved entries due to the element-wise function $P_\Omega(\cdot)$. Third, the adjacency matrix $\mathbf{A}_{GCN}$ used in GloGNN is the symmetric normalized adjacency matrix (or its powers) used in vanilla GCN, whose edge weights are uniform and positive. In contrast, we use the pseudo labels to generate a similarity matrix that allows negative weights.

## 5  EXPERIMENT

In this section, we evaluate the performance of LRGNN. We put the experimental results w.r.t. the convergence rate of the softImpute-ALS algorithm, ablation study, robustness to random noise added to the signed adjacency matrix, and a large-scale graph in Appendix A.9 due to space limitation.

**Datasets.** We use three homophilious datasets including Cora, Citeseer and Pubmed (Yang et al., 2016). We also use 6 heterophilious datasets released in (Pei et al., 2020) and (Rozemberczki et al., 2021). The training/validation/testing splits used in this paper are the same as (Pei et al., 2020). The datasets and splits are all available from the Pytorch Geometric library (Fey & Lenssen, 2019). Details of these datasets are provided in Appendix A.5.

**Baselines.** We compare LRGNN with 11 baselines, including (1) classic GNN models: vanilla GCN (Kipf & Welling, 2017), GAT (Velickovic et al., 2018) and MixHop (Abu-El-Haija et al., 2019). (2) GNN models dedicated to tackling heterophily: H$_2$GCN (Zhu et al., 2020), GPR-GNN (Chien et al., 2021), WRGAT (Suresh et al., 2021), LINKX (Lim et al., 2021), GGCN (Yan et al., 2021), ACM-GCN (Luan et al., 2021), and GloGNN++ (Li et al., 2022). (3) 2-layer MLP. We choose GloGNN++ and ACM-GCN as they generally perform better than other variants proposed in the corresponding papers.

**Node classification results.** Table 1 summarizes the test accuracy of the methods on the node classification task over datasets with diverse homophily ratios. We may make several observations from the table. (1) MLP is a strong baseline for heterophilous datasets. It significantly outperforms GCN, GAT, and MixHop on Texas, Wisconsin, and Cornell datasets. For example, the average classification accuracy of MLP on Texas is 80.81% while that of GCN is 55.14%. (2) We observe

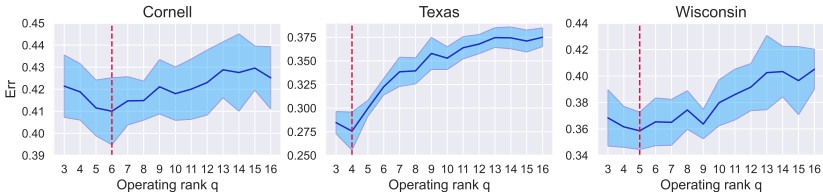

Figure 3: Recovery error of matrix factorization on three datasets. The lowest point is associated with a vertical line. The shaded region corresponds to a 95% confidence interval.

|  | Texas | Wisconsin | Cornell | Actor | Squirrel | Chameleon | Cora | Citeseer | Pubmed |
|---|---|---|---|---|---|---|---|---|---|
| H$_2$GCN | 5.4/22.4 | 6.8/22.7 | 4.5/26.6 | 52.9/14.7 | 22.3/7.2 | 7.0/3.3 | 5.9/2.7 | 9.6/4.9 | 210/201.5 |
| GGCN | **6.8/2.5** | 17.5/7.7 | 24.4/8.2 | 200/54.5 | 54.1/38.2 | 30.2/14.3 | 237/69.6 | 123/20.8 | - |
| ACM-GCN | 15.9/6.7 | **20.2/4.2** | 18.6/6.3 | 18.7/4.9 | 21.8/11.5 | 20.8/9.2 | 20.8/17.3 | 21.3/9.5 | 17.7/14.5 |
| GloGNN++ | 10.1/18.8 | 27.2/62.9 | 15.1/5.8 | 37.9/4.8 | 21.8/5.6 | 19.1/4.9 | 32.8/5.1 | 25.3/11.5 | **19.5/6.6** |
| LRGNN | 11.3/6.0 | 21.2/7.3 | **20.2/4.8** | **17.1/1.5** | 24.9/5.1 | **22.2/4.5** | **17.5/1.9** | **26.3/2.4** | 32.2/8.8 |

Table 2: Empirical running time comparison. Average running time per epoch(in ms)/average total running time(in s). - indicates that the algorithm fails to converge within 10 minutes.

two different patterns among the heterophilious datasets. GCN, GAT, MixHop, and LINKX perform worse than MLP on Texas, Wisconsin, and Cornell. However, they outperform MLP on Squirrel and Chameleon datasets. (3) Generally, methods dedicated to heterophilous graphs perform better than MLP and traditional GNNs. GPR-GNN is the exception. (4) H$_2$GCN (6.22), WRGAT (5.78), GGCN (4.00), ACM-GCN (3.56), GloGNN++ (2.67), and LRGNN (1.56) are the top-performers (the number in parentheses corresponds to the average rank across all datasets). LRGNN performs the best in terms of average rank. This shows that LRGNN can consistently offer superior performance on both homophilious and heterophilious graphs. Notably, LRGNN achieves the best result on Squirrel with around 20.5% improvement over the runner-up score achieved by LINKX. It is worth noting that the superior performance of LRGNN on Squirrel and Chameleon may be attributed to the relatively large average node degrees of these two datasets (38.16 and 13.8, respectively), which indicates that we have more observations to recover the underlying complete graph.

**Choice of operating rank $q$.** We investigate how the operating rank $q$ affects the performance. Specifically, we are interested in the recovery loss of MF: $err = \frac{1}{2n^2} \sum_{i,j} |sign((\mathbf{U}_*^{(L)} \mathbf{V}_*^{(L)^T})_{i,j}) - sign(\mathbf{A}_{i,j}^*)|$, where $L$ denotes the last layer. From Figure 3 we can observe that as $q$ increases, the error gradually decreases at first, then the error gradually rises after the lowest point (associated with a vertical line). For these three datasets, the lowest points are 6, 4, and 5, respectively. Note that these datasets all have 5 classes while for Texas, there is one class that only contains one node. This result empirically verifies the low-rank structures of real-world graphs. We can conclude that the best choice of operating rank $q$ should be exactly the number of classes or slightly larger than it. This also explains the effectiveness of low-rank modeling: we can explicitly choose a proper rank for the coefficient matrix according to the number of classes since in most cases, the factors $\mathbf{U}$ and $\mathbf{V}$ are full rank matrices, so the rank of the coefficient matrix is exactly $q$. In general LRMF problem, we have no exact information about the rank of the matrix we want to recover. However, we do know the rank when considering node classification tasks, which makes choosing a proper $q$ easier.

**Results on synthetic graphs.** To comprehensively evaluate the performance of LRGNN, we use random partition graphs (Kim & Oh, 2021) generated by stochastic block model. The node features are sampled from Gaussian distributions where the centers of clusters are vertices of a hypercube. Note that the distance between the means of Gaussian distributions is small compared to the standard deviation. As a result, node features of different classes are hard to distinguish, which can be verified by the poor performance of MLP on these synthetic graphs. We use 15 synthetic random graphs with varying node-level homophily ratios and average degrees. More details can be found in Appendix A.5. We can make the following observations from Figure 4. When homophily ratio or degree is low, *i.e.*, average degree = 0.5, homophily ratio = 0.1, and homophily ratio = 0.3,

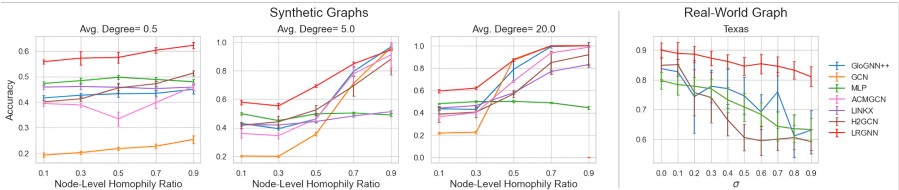

Figure 4: The first three subfigures are results on synthetic graphs and the last one is the result on corrupted Texas. Error bars indicate $95\%$ confidence interval.

LRGNN is the only one that can achieve higher accuracy than MLP. Besides, LRGNN outperforms other methods by a large margin when the homophily ratio and average degree are not too high. We note that LRGNN also significantly outperforms all the baselines on Squirrel and Chameleon. These graphs have a common characteristic, *i.e*, the quality of node features is not good enough for MLPs to achieve an acceptable result. We conjecture that LRGNN performs particularly well on these kinds of graphs, that is, LRGNN can offer superior performance even if the node features are not so informative. We next validate our conjecture using a real-world dataset. Specifically, we degrade the quality of features of Texas dataset and then test the performance of models trained on this corrupted dataset. We add Gaussian variables to the features and obtain a degraded feature matrix by $\mathbf{X}'_{i,j} = \mathbf{X}_{i,j} + \epsilon_j$, with $\epsilon_j$ i.i.d. sampled from a Gaussian distribution $\mathcal{N}(0, \sigma^2)$. Note that the original features will be overwhelmed by the Gaussian random variables if a large $\sigma$ is applied, and thus the features are less informative. We take GloGNN++, $H_2$GCN, and MLP as representative models for comparison. We can observe from Table 4 that, as the features get less informative, the performance of these three methods deteriorates dramatically. For example, when $\sigma = 0.8$, their accuracies are around $60\%$, while the accuracy of LRGNN is above $80\%$. In addition to that, a large $\sigma$ also makes their training process erratic, which is accompanied by significant error bars. These results empirically confirm our previous analysis that LRGNN is particularly effective in the scenario that node features are not that informative. This may explain the effectiveness of LRGNN, namely deriving the coefficient matrix by leveraging both the node representations and observed signed edges.

**Efficiency study.** We also evaluate the efficiency of LRGNN compared to other baseline models. We select GGCN, ACM-GCN, $H_2$GCN, and GloGNN++ for comparison as they are the top performers in Table 1. We exclude WRGAT since it takes a long time to precompute the multi-relational graphs. For GGCN, Glo-GNN++, and $H_2$GCN, we use the codes and hyper-parameters provided by their authors. Since there is no available code for ACM-GCN, we implement it using Pytorch Library and tune the hyper-parameters based on validation set. We can observe from Table 2 that LRGNN has the shortest running time on 6 out of 9 datasets. Especially, LRGNN converges within 10 seconds over all the datasets. As a comparison, GGCN fails to converge within 10 minutes on Pubmed. Besides, LRGNN achieves around 8.5× and 4.8× speedups compared with GloGNN++ on Texas and Citeseer, respectively. In conclusion, LRGNN is efficient and converges very fast. It is worth noting that to achieve the reported results of LRGNN in Table 1, the number of layers and the update iterations for softImpute-ALS algorithm are nearly always 1. This explains the superior efficiency of LRGNN and confirms the fast convergence rate of softImpute-ALS algorithm over the neural-style initialization.

## 6 CONCLUSION

In this paper, we address the challenge of generalizing GNNs to heterophilious graphs with the help of the weak balance theory. Inspired by the low-rank structures of the weakly-balanced graphs, we propose to explicitly model the coefficient matrix as a low-rank matrix. The coefficient matrix is derived by solving a difficult non-convex optimization problem whose objective function consists of an LRR and an LRMF term. Extensive experimental results have demonstrated the effectiveness of the proposed LRGNN, especially when the node features are not informative enough.

