# OpenReview forum: "Low-Rank Graph Neural Networks Inspired by the Weak-balance Theory in Social Networks"
_ICLR.cc/2023/Conference — Submitted to ICLR 2023_

### Official Review · Reviewer_TShN · 2022-10-24

**Confidence:** 2
**Correctness:** 4
**Technical Novelty And Significance:** 1
**Empirical Novelty And Significance:** Not applicable
**Recommendation:** 3

**Clarity, Quality, Novelty And Reproducibility:**

- The clarity and reproducibility are good.
- The novelty and the originality are weak.

**Strength And Weaknesses:**

Strength
- The motivation of low-rank on adjacency matrix makes sense.
- The experimental evaluations are sufficient.
- The connection between subspace learning based on LRR and the proposed method is interesting.

Weaknesses
- The novelty of this paper is limited.  The low-rank constraint on adjacency matrix is not novel, such as [1]. This paper just employs the softimpute alternating least square to speedup the computation. The construction of \tilde{A} at the beginning of page 5 is not novel yet. This strategy has been proposed in dealing networks with heterophily, such as [2].
- The organization is redundant.

[1] Wei Jin, Yao Ma, Xiaorui Liu, Xianfeng Tang, Suhang Wang, Jiliang Tang: Graph Structure Learning for Robust Graph Neural Networks. KDD 2020: 66-74

[2] Jiong Zhu, Ryan A. Rossi, Anup Rao, Tung Mai, Nedim Lipka, Nesreen K. Ahmed, Danai Koutra: Graph Neural Networks with Heterophily. AAAI 2021: 11168-11176

**Summary Of The Paper:**

This paper provides a post-processing to the LINKX with the low-rank constraint on the adjacency matrix. Then, it employs softimpute alternating least square to reduce the complexity. Finally, it bridges the connection between the proposed methods and subspace clustering based on low-rank recovery. Experimental evaluations verify its superiority.


**Summary Of The Review:**

My main concern is the limited novelty as shown in weakness.

---

> ### Author Response · Authors · 2022-11-19
> **Response to Reviewer TShN**
>
> Thanks for accknowledging that our motivation makes sense and the experimental evaluations are suffcient. Please find the detailed comments below.
>
> **Q1** The low-rank constraint on adjacency matrix is not novel. This paper just employs the softimpute alternating least square to speedup the computation.
>
> **R**  It should be noted that the motivaton of our paper is sifnificantly different from Li et al., 2020 (KDD 2020) [1] in the aspects of motivation,  ojbective, and methodology. First, motivated from the weak-balance theory of signed networks, we introduce  the methods of low rank matrix factorization to model graphs with heterophily, while  Li et al. utilize the low-rank and sparseness for defense over adversarial attackes which is totally different from our paper. Thus, the research problems to address are totally different, leading to different objective functions, different benchmarks, different evaluation measures, etc. For example, LRGNN considers both negative and positive links and the coefficient matrix is a dense matrix, while Pro-GNN only considers positive links.
>
> Secondly, the low-rank property in LRGNN is guaranteed by matrix factorization term from a optimization problem with closed-form solution, while the low-rank property in Pro-GNN is only encouraged by simply adding a nuclear term to the loss function. Thus, the differences for low-rank constraint between LRGNN and Pro-GNN are not just about "employs the softimpute alternating least square to speedup the computation.".
>
> More importantly, our novelty in the use of low-rank matrix factorization has been clearly acknowledged by other reviewers.
>
> aB3d: "The paper draws from well-established theory on low-rank matrix factorizations and signed networks."
>
> 4e9z:  "S1: The paper has a clear motivation for explicitly exploiting the low-rank property of the signed social graph.""
> "The work is novel in motivation and good in performance"
>
> Yww3: "This paper proposed a novel solution using low-rank matrix to the problem of GNN learning on graph with heterophily. "
>
>
> **Q2** The construction of \tilde{A} at the beginning of page 5 is not novel yet. This strategy has been proposed in dealing networks with heterophily.
>
> **R2**
> Thanks for providing [2] and we have added the reference in our revision.
> Please also note that there is a significant difference.  We use the pseudo labels to construct a signed adjacency matrix $\tilde{A}$ for matrix factorization, and the use of signs is crucial which brigdes the connection between signed networks and heterophilious graphs; While Jin et al. [2] use the pseudo labels to build a compatibility matrix for propagation.
>
> **Q3** The organization is redundant
>
> **R3** Given no detailed description, we are not clear about the meaning of this comment.

---

### Official Review · Reviewer_Yww3 · 2022-10-24

**Confidence:** 3
**Correctness:** 4
**Technical Novelty And Significance:** 3
**Empirical Novelty And Significance:** 3
**Recommendation:** 5

**Clarity, Quality, Novelty And Reproducibility:**

The paper describes the proposed algorithm with great clarity and the algorithm itself is novel as it is based on a novel interpretation of the coefficient matrix.

But still, there are few issues that prevent me from understanding this paper:
1) The paper claims "signed graph are naturally generalized c-weakly ... task with c classes." After reading the paper, I still find it confusing. Are the edges here defined as whether two nodes belong to the same classes? If so, then it is trivial. How does this justify the low rank assumption of the coefficient matrix?
2) The derivation of the formula (19) is based on the approximation of the (18). I think the approximation step should be provided and an analysis of the approximation error is also quite helpful. Dropping the nuclear norm term could have non-trivial impacts to the final solutions.
3) The objective for the GloGNN is different from the paper which uses k-hop $A_{gcn}$.


**Strength And Weaknesses:**

Strength:
1) The paper connects the representation learning on the graph with heterophily and the signed networks. This connection has not been explored by previous literatures and might provide more insights of the learned representations.
2) The motivations and designs of the algorithms are articulated clearly with both theoretical derivations and numerical simulations.
3) The authors conduct extensive experiments to verify the effectiveness of the algorithm and provides ablation study and diagnostic plots to verify their modeling assumptions.

Weaknesses:
1) Although novel, the connection between signed networks and the modeling target is not clear. And thus, the motivation of assuming low-rank approximation needs further elaborations.
2) The graphs used in the experiments are not large. Does this proposed solution scale to graphs with hundreds of thousands of nodes?


**Summary Of The Paper:**

This paper proposes a low-rank matrix to approximate the coefficient matrix which describes the aggregations of neighborhood information globally. The paper designed a new coefficient matrix with the form of a low rank matrix $UV^t$ which is the solution of the minimization problem inspired by the subspace clustering problem. Furthermore, author proposes a semi-supervised pseudo label generation process as well as matrix initialization step using classic neural approaches on graphs. To solve this matrix, author proposes a novel alternating update algorithms based on softimpute and proves its correctness. The proposed solution achieves results on real-world datasets comparable to other state of the art models and consistently outperforms other methods on synthetic datasets.

**Summary Of The Review:**

This paper proposed a novel solution using low-rank matrix to the problem of GNN learning on graph with heterophily. This paper is in general well written and provide a new perspective. However, the experimentation results are not very strong and the scalability of the solution is not verified. Besides, although the author mentions the connection between signed networks and the heterophilous graphs, it is not clear to me whether there is non-trivial connection. Considering this is the core concept and motivation, I am currently giving a reject but I am willing to raise my score if the authors could provide additional explanations.

---

> ### Author Response · Authors · 2022-11-19
> **Response to Reviewer Yww3**
>
> Thanks for your suggestions and here are our response to your questions.
>
> **Q1** The paper claims "signed graph are naturally generalized c-weakly ... task with c classes." After reading the paper, I still find it confusing & the connection between signed networks and the modeling target is not clear.
>
>
> **R:*** Our basic idea is to recover a ground-truth signed coefficient matrix, which is defined as $A_{i,j}=1$, if node $v_i$ and $v_j$ belong to the same class and $A_{i,j}=-1$ otherwise. Then we can use this coefficient matrix to propagation node features:
> $H^{l+1} = AH^{l}$. Ideally, we could achieve perfect node classification if we perfectly recover this coefficient matrix.
>
> This target coefficient matrix is low-rank, which is the direct result of the following observation:
> there are only c (the number of classes) distinct rows in the matrix.  For example, if $v_i$ and $v_k$ belong to the same class, they share the same set of positive edges and the same set of negative edges, which means that $A_{i,:}$ and $A_{k,:}$ are equal. This is because by definition, $A_{i,j}=A_{k,j}$, for any $v_j$. When c>2, they are linearly independent. Therefore, the rank of this matrix is c if c>2 and 1 if c=2.
> The low-rank structure of the coefficient matrix implies that we can recover it using low-rank matrix factorization, which is both scalable and effective.
>
> In summary, we are using low-rank approximation for three reasons:
> (1) Signed social networks are greatly similar to heterophilious graphs: they both consider the use of negative links and both follow some common patterns in structure[1].
> (2) The coefficient matrices of a signed social networks are low-rank.
> (3) Low-rank approximation for link prediction in signed social networks is a representative classical method which has achieved significant successes [2],[3].
>
>
> **Q2** The graphs used in the experiments are not large. Does this proposed solution scale to graphs with hundreds of thousands of nodes?
>
> **R2** Yes, we have experimented on some large-scale datasets. In addition to the results of Penn94 reported in the supplementary matters in previous submission, we list  the results regarding efficiency (seconds per epoch) and performance on additional large-scale datasets (the remaining datasets released in the LINKX paper are too large to run on our machine with a 16GB GPU).
>
> |Model|Penn94 |arXiv-year|genius|
> |:---:|:---:|:---:|:---:|
> |LRGNN|**86.71**|55.94|**91.10**|
> |GloGNN++|85.74|54.79|90.91|
> |LINKX|84.71|**56.00**|90.77|
> |ACM-GCN|82.52|47.37|80.33|
> |H2GCN|81.31|OOM|OOM|
>
> |Model|Penn94 |arXiv-year|genius|
> |:---:|:---:|:---:|:---:|
> |LRGNN|0.34(s)|0.44(s)|0.64(s)|
> |GloGNN++|0.48(s)|0.33(s)|0.88(s)|
> |LINKX|0.31(s)|0.42(s)|0.66(s)|
> |ACM-GCN|0.22(s)|0.22(s)|0.42(s)|
> |H2GCN|1.1(s)|OOM|OOM|
>
> Overall, LRGNN has a running time and node classification accuracy comparable to GloGNN++ and LINKX.
>
>
>
> **Q2** "The derivation of the formula (19) is based on the approximation of the (18)..."
>
> **R2** Good question. According to recent research[4][5], the soluton of the matrix factorization problem is also a solution to the nuclear norm minimization problem, when the number of columns of $\mathbf{U}$ (i.e., $q$) is larger than the optimal rank of $\mathbf{Z}^*$.
>  Therefore, by properly selecting a sufficient large $q$ with $q \ge c$, (19)  is equivalent to (18).
>
> **Q3** "The objective for the GloGNN is different from the paper which uses k-hop $A_{GCN}$."
>
> **R:** Thank you for your careful checking. We will add the superscript to the term $A_{GCN}$ in our revised version.
>
> [1]Tang J, Chang Y, Aggarwal C, et al. A survey of signed network mining in social media[J]. ACM Computing Surveys (CSUR), 2016, 49(3): 1-37.
>
> [2]Hsieh C J, Chiang K Y, Dhillon I S. Low rank modeling of signed networks[C]//Proceedings of the 18th ACM SIGKDD international conference on Knowledge discovery and data mining. 2012: 507-515.
>
> [3] Chiang K Y, Hsieh C J, Natarajan N, et al. Prediction and clustering in signed networks: a local to global perspective[J]. The Journal of Machine Learning Research, 2014, 15(1): 1177-1213.
>
> [4] Cabral R, De la Torre F, Costeira J P, et al. Unifying nuclear norm and bilinear factorization approaches for low-rank matrix decomposition[C]//Proceedings of the IEEE international conference on computer vision. 2013: 2488-2495.
>
> [5] Hastie T, Mazumder R, Lee J D, et al. Matrix completion and low-rank SVD via fast alternating least squares[J]. The Journal of Machine Learning Research, 2015, 16(1): 3367-3402.

---

> > ### Comment · Reviewer_Yww3 · 2022-12-05
> > **Need more evidences to support the low rank assumptions**
> >
> > Thanks the authors for providing the additional results. The answers to q2 and s3 are satisfactory. However, the answer to q1 is not. It is not non-trivial that the coefficient matrix is low-rank if it is defined based on node classes. The core issue is why we want the propagation matrix to be low-rank. "Ideally, we could achieve perfect node classification if we perfectly recover this coefficient matrix". Does this imply that the optimal propagation matrix for GNN should be low-rank? If so, there is an easy counter example: if add a C * eye(N) (C is sufficiently large) matrix to the ground-truth signed coefficient matrix to make it diagonal dominant, then it is full rank. In general, I expect further elaborations about this connection in the main paper.

---

> > > ### Author Response · Authors · 2022-12-07
> > > **Explanations for low-rank assumptions**
> > >
> > > Thanks for your comments.
> > >
> > > **Q**
> > >
> > > The core issue is why we want the propagation matrix to be low-rank.
> > >
> > > **R**
> > >
> > > Let $A^*$ denote the ground truth coefficient matrix, $N$ be an arbitrary matrix has the same shape with $A^*$ . Then $A^*+N$ would also be an optimal propagation matrix  if $sign(A^*+N) = sign(A^*)$, where $sign(\cdot)$ is an element-wise function. Hence, the full-rank matrix you mentioned is definitely an optimal propagation matrix and the optimal propagation matrix is **not** necessarily a low-rank matrix. However,
> > >
> > > (1) Theoretically, we choose to recover $A^*$ but not $A^*+N$ because recovering $A^*+N$ is **not** always possible in general, especially when $A^*+N$ is not low-rank[1]. The number of observations needed for recovery is linear to the rank of $A^*+N$. According to the theoretical results in [2], the required number of observations $m$ for perfect recovery is bounded as $m \ge Cn^{1.2}rlog_{n}$, where $r$ is the rank of the matrix. If we choose to recover $A^*+C*eye(N)$, we need more than $n^2$ observation ($r=n$), which is impossible!  Therefore, it is better to select the ground truth coefficient matrix to recover since it has the lowest rank among all the optimal propagation matrices ($r=c$, a very small number that can be neglected).
> > >
> > > (2) Empirically, the experiment regarding the choice of the operating rank in Sec.5 has demonstrated that the best performance for recovery is achieved when $U$ and $V$ have rank $c$, and the performance degenerates as the rank increasing. Besides, the time and space complexities of the algorithm are also linear to the rank of the matrix (see time complexity analysis in Appendix A.3). Hence, using a non low-rank $A^*+N$ would induce more computational costs.
> > >
> > > In conclusion, we want the propagation matrix to be low-rank not because $A^*$ is the only optimal propagation matrix, but because it is the easiest and cheapest to recover among all the optimal propagation matrices.
> > >
> > > We hope our response can address your concerns. Please let us know if you have further question!
> > >
> > > [1] Davenport, Mark A., and Justin Romberg. "An overview of low-rank matrix recovery from incomplete observations." IEEE Journal of Selected Topics in Signal Processing 10.4 (2016): 608-622.
> > >
> > > [2] Candes, Emmanuel, and Benjamin Recht. "Exact matrix completion via convex optimization." Communications of the ACM 55.6 (2012): 111-119.

---

### Official Review · Reviewer_4e9z · 2022-10-25

**Confidence:** 5
**Correctness:** 2
**Technical Novelty And Significance:** 2
**Empirical Novelty And Significance:** 2
**Recommendation:** 3

**Clarity, Quality, Novelty And Reproducibility:**

- Clarity: The paper is overall friendly to readers.
- Quality and Novelty: The work's motivation is clear, but it is largely incremental compared to previous work.
- Reproducibility: The code is released.


**Strength And Weaknesses:**

Strengths:

S1: The paper has a clear motivation for explicitly exploiting the low-rank property of the signed social graph.

S2: This paper achieves good performance on node classification tasks of heterophilic graphs.

S3: Figure 3 shows that, as the operating rank of U and V increases, the recovery error for signed matrix first decreases, and then increases, which is consistent with the weak balance theory.

S4: LWGNN inherits the satisfactory time complexity from GloGNN. The authors prove that the time complexity of the forward propagation process is linear to the edge number, which is commendable considering that the process of finding the closed-form solution involves the computation of dense matrix multiplication and inversion.

Weaknesses:

W1: The model largely inherits GloGNN in each component, which limits the contribution of this work.

W2: Many practices in this paper are borrowed elsewhere without a clear claim, e.g.,
- Equation (5), MLP(A) in H^{(0)} is similar to the approach in LINKX;
- Equation (6), the initial residual connection is similar to APPNP;
- The first term (propagation term) in Equation (7) is from GloGNN.
These are all powerful submodules. The paper should indicate the source of the design ideas. Also, ablation experiments for these modules are encouraged.

W3: Due to the integration of several modules, it is not clear if the utility of signs is crucial for performance gain. The \tilde{A} in Equation 7 (i.e., the set of edge signs with pseudo labels) should be replaced by the vanilla adjacency matrix A to verify that the supervision of edge sign plays a role. Note that the ablation model is not the same as GloGNN.

W4: Still, due to the integration of several modules, the model has more hyperparameters. For example, \mu from the use of MLP(A), \delta from the use of pseudo-labeling, which may lead to a decrease in the usefulness of the model.

W5: The proof in Appendix 1 seems to be incomplete.

W6: Despite S4, the paper is only experimental on smaller (N<20,000). Thus, experiments on the arxiv-ogbn dataset or some LINKX datasets to highlight S3 might be beneficial.


**Summary Of The Paper:**

This paper proposes Low-Rank Graph Neural Network (LWGNN), which enhances GNNs by utilizing low-rank approximation to recover the underlying fully-connected matrix Z. In the matrix Z, a positive Z[i,j] means that node j and node i belong to the same class, and a larger value of Z[i,j] means that node j has more influence on node i and vice versa. By using Z as the propagation matrix, LWGNN achieves good performance on node classification tasks of heterophilic graphs.

The motivation for the low-rank representation comes from the weak balance theory in Signed Social Networks, which leads to a low-rank property of the underlying fully connected signed graph obtained from the classes of each node. Thus, the authors represent Z as the product of two low-rank matrices U and V. For each layer, the authors obtain U, V by (1). reasonable optimization objectives (Equation 7) with closed-form solutions (Equation 9-12), i.e. softImpute-ALS algorithm; (2). carefully designed initialization (Equation 14); and (3). pseudo-labeling (Equation 8) to fulfill the objective function.

This paper is an extension to GloGNN [1] in that they both require solving the propagation matrix Z in each layer, and using it for propagation. The main differences between LWGNN and GloGNN are: 1) LWGNN makes explicit use of low-rank matrix and low-rank approximation, while GloGNN only implicitly uses low-rank matrices for computing acceleration; 2) LWGNN draws the Z close to the signed adjacency matrix A, while GloGNN does not utilize the signs.


**Summary Of The Review:**

The work is novel in motivation and good in performance. However, it is largely built on existing work, and it’s not very clear which components its power gains from.

---

> ### Author Response · Authors · 2022-11-19
> **Response to Reviewer 4e9z**
>
> Thanks for your comments and we have carefully revised our paper according to the comments.
>
> **Response to W1, W2, & W3**
>
> **R:** We appreciate the great advances in GNNs, including LINKX, APPNP, GloGNN, etc, which provide many inspirations and form the foundations of our research. We have revised our paper with more appropriate acknowledgments. It is important to note that the first term of Eq (7) is indeed derived from the LRR method (Liu et al., 2010), as stated in Section 4. We noticed the connection to GloGNN (described in Eq(20) ) and discussed their significant differences in Section 4 in our previous submission. In particular, the solutions given by GloGNN are nearly full-rank (note that the coefficient matrix of GloGNN is decomposed into the sum of many low-rank matrices, which is no longer low-rank).  Full-rank coefficient matrix is sub-optimal as evidenced by our experiments on the recovery error. We provide the ranks of coefficient matrices given by GloGNN++ and LRGNN in the following table.
>
> |Model|Texas |Cornell|Wisconsin|
> |:---:|:---:|:---:|:---:|
> |LRGNN|10|6|6|
> |GloGNN++|182|180|250|
>
> Especially, we want to emphasize the contribution of this paper --- the idea of explicitly exploiting the low-rank property of the signed social graph, which are not explored in these studies, has greatly improved the performance of node classification on both homophiles and heterophiles graphs .
>
>
> Additionally, we have conducted a more comprehensive ablation study to shed light on the contribution of the components. In the following table, LRGNN-Uni indicates that the signed adjacency matrix is replaced with the uniform sparse adjacency matrix in GCN; LRGNN-Reg indicates that the matrix factorization term is replaced with a Frobenius term; LRGNN-DA drops the MLP(A) from $H^{(0)}$.
>
>
> | Model | Texas | Wisconsin | Cornell | Actor| Squirrel | Chameleon | Cora| Citeseer | Pubmed|
> | :---: | :---:| :---:| :---:| :---:| :---:| :---:| :---:| :---:| :---:|
> |LRGNN|**89.19**| **88.23**|**86.22**|**37.10**|**74.51**|**78.93**|**88.23**|**77.46**|**89.60**|
> |LRGNN-Uni|87.84| 82.94|85.14|34.53|69.49|68.46|87.67|**77.29**|89.39|
> |LRGNN-Reg|88.38| 82.94|83.51|36.61|65.83|69.12|86.92|75.57|87.36|
> |LRGNN-DA|**89.19**| 86.86|85.95|36.86|72.52|75.65|**88.23**|77.32|89.45|
>
> The results demonstrate that the uses of both MF and signs are crucial, dropping these components causes significant degradations in performance.
>
> **W4 More hyper-parameters**
>
> **R:** For fair comparison, we have restricted the search space for hyper-parameters. For example, #layers is fixed at 1; #iterations is fixed at 1 to 2. More details can be found in appendix A.10.
>
> **W5 The proof in Appendix 1 seems to be incomplete.**
>
> **R:** Appendix 1 aims to disclose the reason to employ the softimpute algorithm, that is, dropping the element-wise function such that we can directly derive the closed-form solution.
>
> **W6 Experiments on LINKX datasets**
>
> **R:** We indeed put the results on Penn94 (N = 41,554) in the supplemenary material (A.9) in the previous submission. We provide more results regarding performance and efficiency (seconds per epoch) on large-scale datasets.
>
> |Model|Penn94 |arXiv-year|genius|
> |:---:|:---:|:---:|:---:|
> |LRGNN|**86.71**|55.94|**91.10**|
> |GloGNN++|85.74|54.79|90.91|
> |LINKX|84.71|**56.00**|90.77|
> |ACM-GCN|82.52|47.37|80.33|
> |H2GCN|81.31|OOM|OOM|
>
> |Model|Penn94 |arXiv-year|genius|
> |:---:|:---:|:---:|:---:|
> |LRGNN|0.34(s)|0.44(s)|0.64(s)|
> |GloGNN++|0.48(s)|0.33(s)|0.88(s)|
> |LINKX|0.31(s)|0.42(s)|0.66(s)|
> |ACM-GCN|0.22(s)|0.22(s)|0.42(s)|
> |H2GCN|1.1(s)|OOM|OOM|

---

### Official Review · Reviewer_aB3d · 2022-10-26

**Confidence:** 2
**Correctness:** 4
**Technical Novelty And Significance:** 3
**Empirical Novelty And Significance:** 4
**Recommendation:** 6

**Clarity, Quality, Novelty And Reproducibility:**

The paper is well-written and easy to follow. It presents a novel GNN type which takes a different approach from existing convolutional architectures.

**Strength And Weaknesses:**

Strengths
-

- The paper draws from well-established theory on low-rank matrix factorizations and signed networks. It is able to adapt an existing ALS algorithm to a new case using a surrogate loss term, which it justifies using a bound. There is a nice discussion of computational complexity and methods to speed up the optimization.
- LRGNNs significantly outperform other types of GNNs empirically on the heterophilic datasets, especially Squirrel.
- The experiments are comprehensive and provide good intuition as to why LRGNNs are able to perform well. Adding node-wise independent noise to each of the feature vectors hurts most GNNs, but LRGNNs are robust to this.

Weaknesses
-

- All of the datasets used seem to rely only on short-range interactions; this is evidenced by each GNN only using one layer for each of the datasets. This brings up the question of how much the graph structure is actually being used. How would LRGNNs compare in the cases where deeper GNNs are necessary?

**Summary Of The Paper:**

The paper proposes a new GNN architecture inspired by a model of homophily and heterophily in social networks. Nodes with similar features are modeled with a positive edge, while nodes with different features are modeled with a negative edge. The paper notices that the weak balance theory for networks naturally applies in the setting of GNNs, since the nodes can be grouped according to which class they belong to. Using existing theory of weakly balanced networks being of low rank, the paper uses a GNN model which explicitly tries to compute a low-rank signed coefficient matrix for node aggregation. This architecture is evaluated for several node classification tasks and compared to existing GNNs.

**Summary Of The Review:**

I believe that the contributions of the paper are significant and are able to handle heterophily well, which other GNNs have struggled with.

---

> ### Author Response · Authors · 2022-11-19
> **Response to Reviewer aB3d**
>
> Thanks for your constructive comments and here are our response to your questions.
>
> **Q1** "All of the datasets used seem to rely only on short-range interactions. this is evidenced by each GNN only using one layer for each of the datasets..."
>
> **R1** In fact, long-range interactions are beneficial to the node classification task on these datasets.  This is partially because high-order neighbors could contain more homophilious nodes. For example, [1] proves that under certain assumptions, two-hop neighbors are homophily-dominant.
>
> We think the main advantages of deep GNNs come from the power that enables nodes to aggregate features from distant nodes. However, in our model design, $\mathbf{UV}^{T}$ is a global coefficient matrix which means that a 1-layer LRGNN can already utilize global information, that is, each node can aggregate features from all the other nodes. Therefore, adding more layers has improvement in this aspect and induces more computation costs.
> Here are the results of 2-layer LRGNN (LRGNN-2):
>
> | Model | Texas | Wisconsin | Cornell | Actor| Squirrel | Chameleon | Cora| Citeseer | Pubmed|
> | :---: | :---:| :---:| :---:| :---:| :---:| :---:| :---:| :---:| :---:|
> |LRGNN|**89.19**| **88.23**|**86.22**|**37.10**|**74.51**|**78.93**|**88.23**|77.46|**89.60**|
> |LRGNN-2|88.12| 86.86|86.22|36.58|71.35|77.24|87.91|**77.51**|88.32|
>
> [1] Zhu J, Yan Y, Zhao L, et al. Beyond homophily in graph neural networks: Current limitations and effective designs[J]. Advances in Neural Information Processing Systems, 2020, 33: 7793-7804.

---

### Public Comment · ~Benedek_Andras_Rozemberczki1 · 2022-11-05
**Misattribution of datasets**

The paper misattributed the authorship of the Chameleons and Squirrels datasets. These datasets were proposed in this ICLR submission:

https://openreview.net/forum?id=HJxiMAVtPH

The Pei et al. paper cited by the authors took the Squirrel and Chameleons datasets and used those for benchmarking, but had nothing to do with the creation of the datasets. The correct citation for the paper which proposed the datasets is:

```bibtex
>@article{musae,
          author = {Rozemberczki, Benedek and Allen, Carl and Sarkar, Rik},
          title = {{Multi-Scale Attributed Node Embedding}},
          journal = {Journal of Complex Networks},
          volume = {9},
          number = {2},
          year = {2021},
}
```

---

> ### Author Response · Authors · 2022-11-19
> **Hi Benedek,**
>
> Thanks for pointing this out. We have cited your paper in our revised version.
>
> Best,
>
> Authors

---

### Author Response · Authors · 2022-12-01
**Further questions and comments**

We thank the reviewers for spending the time and effort to review our paper and provide constructive suggestions. We have improved our paper based on the comments. We hope that our response addresses your concerns. Since we are reaching the end of the discussion period, please let us know if there are any further questions or comments we can address.

---

### Decision · Program_Chairs · 2023-01-20

**Decision:**

Reject

**Justification For Why Not Higher Score:**

The reviewers point out several limitations of the current paper which I agree with. My main concern is the ad-hoc way of the paper to aggregate multiple components and optimization procedures into a single framework, which is not elegant in my opinion and hurts the validation of the main idea of the paper.

**Justification For Why Not Lower Score:**

N/A

**Metareview: Summary, Strengths And Weaknesses:**

This paper studies the heterophilious graph problem by introducing the weak-balance theory in signed social networks. Specifically, nodes belonging to the same class are connected, and otherwise disconnected, which results in a low-rank propagation matrix Z that can be decomposed to the product of two low rank matrices U and V. The experimental results are good. However, as suggested by multiple reviewers, 1) the large similarity to existing work GloGNN limits the contribution, 2) there are many components in the model design, which are combined in an ad-hoc way and introduce too many hyperparameters, hurting the integrity of the model and making it hard to justify whether the main motivation of the paper is effective, and 3) the presentation is cluttered, with related works, theorems, equations, and methods presented in a mixed way, where an overall figure illustration will be much helpful.